# Metabolomic Profiling of End-Stage Heart Failure Secondary to Chronic Chagas Cardiomyopathy

**DOI:** 10.3390/ijms231810456

**Published:** 2022-09-09

**Authors:** Martha Lucía Díaz, Karl Burgess, Richard Burchmore, María Adelaida Gómez, Sergio Alejandro Gómez-Ochoa, Luis Eduardo Echeverría, Carlos Morillo, Clara Isabel González

**Affiliations:** 1Grupo de Inmunología y Epidemiología Molecular, GIEM, Escuela de Microbiología, Facultad de Salud, Universidad Industrial de Santander, Bucaramanga 680002, Colombia; 2Glasgow Polyomics, University of Glasgow, Glasgow G12 8QQ, UK; 3Centro Internacional de Entrenamiento e Investigaciones Médicas-CIDEIM, Cali 760031, Colombia; 4Universidad Icesi, Cali 760031, Colombia; 5Clínica de Falla Cardíaca, Fundación Cardiovascular de Colombia, Floridablanca 681004, Colombia; 6Department of Cardiac Sciences, Cumming School of Medicine, Libin Cardiovascular Institute, University of Calgary, Calgary, AB T2N 1N4, Canada

**Keywords:** heart failure, Chronic Chagas cardiomyopathy, metabolomics, metabolites

## Abstract

Chronic Chagas cardiomyopathy (CCC) is the most frequent and severe clinical form of chronic Chagas disease, representing one of the leading causes of morbidity and mortality in Latin America, and a growing global public health problem. There is currently no approved treatment for CCC; however, omics technologies have enabled significant progress to be made in the search for new therapeutic targets. The metabolic alterations associated with pathogenic mechanisms of CCC and their relationship to cellular and immunopathogenic processes in cardiac tissue remain largely unknown. This exploratory study aimed to evaluate the potential underlying pathogenic mechanisms in the failing myocardium of patients with end-stage heart failure (ESHF) secondary to CCC by applying an untargeted metabolomic profiling approach. Cardiac tissue samples from the left ventricle of patients with ESHF of CCC etiology (*n* = 7) and healthy donors (*n* = 7) were analyzed using liquid chromatography-mass spectrometry. Metabolite profiles showed altered branched-chain amino acid and acylcarnitine levels, decreased fatty acid uptake and oxidation, increased activity of the pentose phosphate pathway, dysregulation of the TCA cycle, and alterations in critical cellular antioxidant systems. These findings suggest processes of energy deficit, alterations in substrate availability, and enhanced production of reactive oxygen species in the affected myocardium. This profile potentially contributes to the development and maintenance of a chronic inflammatory state that leads to progression and severity of CCC. Further studies involving larger sample sizes and comparisons with heart failure patients without CCC are needed to validate these results, opening an avenue to investigate new therapeutic approaches for the treatment and prevention of progression of this unique and severe cardiomyopathy.

## 1. Introduction

Chagas disease (CD) is a neglected tropical disease caused by the protozoan parasite *Trypanosoma cruzi* (*T. cruzi)* that affects 6–7 million people in Latin America and more than 300,000 individuals in the United States and Europe; thus, it has been recognized as the parasitic disease with the highest disease burden in the world [1,2]. Chronic Chagas cardiomyopathy (CCC) represents the most common form of chronic organ involvement in CD, developing in around 30% of infected patients and presenting a significantly worse prognosis than other etiologies of heart failure (HF) [3,4]. CCC is characterized by extensive tissue remodeling with histological findings of focal chronic myocarditis and interstitial fibrosis, reflected in a rapidly progressing dilated cardiomyopathy with unique features such as fatal ventricular arrhythmias and ventricular aneurysms [4]. At present, the treatment of CCC is only symptomatic due to the absence of effective pharmacological therapies to treat patients progressing to heart failure, as current evidence does not suggest a benefit of antitrypanosomal agents in patients with established cardiomyopathy [5,6]. Among the therapeutic alternatives, implantable devices such as cardioverter–defibrillators are indicated in patients with high arrhythmogenicity to avoid sudden death [7]. However, in the most severe cases, implantation of left ventricular assist devices or heart transplantation remains the only available therapeutic option for these patients [8,9].

The pathogenesis of CCC is complex and not yet fully understood [4], but involves parasite-driven immune reactions, autoreactivity triggered by parasite persistence, neurogenic alterations, and disorders of the coronary microvasculature proposed as the main pathophysiological mechanisms underlying the disease [4,10,11]. However, recent evidence has shown that during the chronic phase, an exacerbated inflammatory response is the main determinant of myocardial tissue injury [12]. According to autopsy studies, the hearts of patients with CCC exhibit an intense mononuclear inflammatory infiltrate with a multifocal distribution, generating an inflammatory environment characterized by the production of cytokines such as interferon-γ (INF-γ) and tumor necrosis factor-α (TNF-α), and the activation of cytotoxic mechanisms involving CD8+ T cells [13,14,15]. In this context, there is growing evidence linking several inflammatory and metabolic pathways in the setting of chronic inflammation and specifically during the acute phase of CD [16]. However, the metabolic derangements in the human myocardium associated with CCC pathophysiology remain largely unknown.

Mass spectrometry-based metabolomics provides a useful approach for the identification of metabolites in biofluids or tissues and allows the characterization of molecular phenotypes in health and disease. Findings in humans and animal models provide clear evidence indicating severe metabolic alterations in the failing heart [17]. Therefore, the application of metabolomics has the potential to provide insights into pathophysiological interactions of metabolites in Chronic Chagas Cardiomyopathy (CCM); nevertheless, no studies evaluating the metabolomic profile of CCM in human samples have been published [18]. Thus, this exploratory study aimed to evaluate the potential underlying pathogenic mechanisms of the failing myocardium in end-stage heart failure (ESHF) secondary to CCC compared to healthy donor samples using an untargeted metabolomic profiling approach.

## 2. Results

### 2.1. Demographic and Clinical Characteristics

A total of fourteen cardiac tissue samples from patients with ESHF of CCC etiology (*n* = 7) and healthy control donors (*n* = 7) were included in the analysis. All participants were of mestizo ethnicity, with a large predominance of males in both groups (85.7% in both). Patients in the CCC group were significantly older compared to healthy controls (median age: 51 (Q1: 42; Q3: 55) vs. 30 (Q1: 20; Q3: 33) years, respectively, *p* < 0.001). All patients with CCC exhibited severely reduced ejection fractions (<20%) (median 10% (Q1: 10; Q3: 15)) and a deteriorated New York Heart Association (NYHA) functional class. Regarding pharmacological treatment, all CCC patients were receiving beta-blockers, diuretics, intravenous inotropic agents, and immunosuppressive therapy before heart transplantation, while four were receiving anticoagulants and one statins. Moreover, three patients from the CCC group had previously been implanted with a left ventricular assist device. Finally, all healthy donors died from head trauma in the absence of concomitant thoracoabdominal trauma.

### 2.2. Histopathologic Findings

Microscopic evaluation of the left ventricle in the CCC group revealed a process of chronic myocarditis, characterized by an abundant inflammatory infiltrate of mononuclear predominance with a diffuse, nonuniform distribution of lymphocytes, histiocytes, and eosinophils (Figure 1A). Two main modifications were found in the viable myocardial fibers: hypertrophy and elongation. Hypertrophy was defined as an increase in the diameter of the cardiomyocyte cytoplasm with large and irregular nuclei and the presence of lipofuscin in the nuclear poles (Figure 1B). Also, marked thickening and intense hypertrophy of cardiomyocytes characterized by bulky nuclei and increased diameter of the fibers were observed (Figure 1B). These histopathologic findings were exclusive to the CCC group, as the analyses of the control group samples showed normal findings in all cases (Figure 1C).

### 2.3. Metabolomic Profiling in Myocardial Tissue from CCC Patients and Healthy Donors

A total of 1710 annotated metabolites composed the metabolomic profiles of the cardiac tissues evaluated. After removing redundancy between samples, the final number of annotated metabolites was 690. Principal Components Analysis (PCA) analysis showed significant differences in the metabolic profiles from CCC and healthy tissue samples, in which 56.3% of the variance was explained by principal components 1 and 2 (Figure 2).

Among the features, the abundance of 222 metabolites was significantly different between the groups, of which 40 metabolites matched known standards, being categorized as “MSI category 1 [19].” The differentially expressed metabolites mainly consisted of amino acids, carbohydrates, lipids, and nucleosides, with a significant decrease in the metabolite abundance in the myocardial tissue of CCC patients compared to healthy donors (Table 1 and Figure 3).

### 2.4. Functional Pathway Analyses

Metabolite Set Enrichment Analysis (MSEA) was used to explore, identify and interpret the differential patterns of metabolite abundances in CCC patients and healthy controls. This approach showed that the most significantly enriched pathways were the ones related to the metabolism of aspartate, the urea cycle, arginine and proline metabolism, carnitine synthesis, and the Warburg effect (Figure 4A). Complementarily, the functional analysis revealed that the pathways related to alanine, aspartate, and glutamate metabolism, aminoacyl-tRNA biosynthesis, arginine and proline metabolism, glycerophospholipid metabolism, and pantothenate and CoA biosynthesis were the most relevantly altered in the CCC group compared to the healthy controls (Figure 4B). Figure 5 integrates the different molecular pathways with significant differences between the two groups studied.

## 3. Discussion

CCC is a prevalent and complex disease governed by multiple molecular mechanisms for which no effective therapy is currently available [3,20,21,22]. Several studies have performed metabolomic analyses in murine models of CD in an attempt to elucidate cardiac pathophysiological changes during the acute and chronic stages of infection, providing valuable information on the pathways potentially implicated in the disease and the spatial distribution of the alterations [16,23,24]. However, these analyses continue to pose several challenges, highlighting the interspecies differences concerning human cardiovascular physiology, which often leads to contradictory results. In this context, human cardiac tissue analysis offers an important biological source but is limited by sample availability and demanding logistics [25].

In this exploratory study, we applied an untargeted metabolomic profiling approach to investigate potential functional changes in ESHF of CCC etiology. This represents the first metabolomics study performed on human tissue samples from patients with CCC. Our analysis revealed significant biochemical alterations in critical metabolic pathways of hearts with CCC compared to healthy controls, highlighting a significant dysregulation in amino acid metabolism, glycerophospholipid metabolism, and antioxidant mechanisms, among others. Notably, the metabolic profiles observed in this study were characterized by decreased abundance of most metabolites, which could indicate imbalances in the metabolic pathways evaluated. Although diverse in function, these pathways are closely related due to shared substrates and dependence on the products of some pathways to feed alternative pathways for energy production (Figure 5). Finally, the metabolic signatures observed in the present study have close similarities with the metabolic profiles reported in previous CD studies [16,24]; however, significant differences in additional pathways critical to the myocardial metabolism are reported for the first time.

### 3.1. Amino Acids as Metabolic Substrates

Despite contributing to a much lesser degree to ATP generation compared with fatty acids and glucose, amino acids (AAs) play critical roles in myocardial processes such as protein synthesis, signaling and metabolic intermediates, redox biology, and calcium homeostasis; however, under conditions of prolonged stress or ischemia, amino acids can become anaplerotic substrates [26]. Particularly, amino acids like aspartate, glutamate, glutamine, asparagine, and the branched-chain amino acids (BCAAs) are preferentially used by the heart as metabolic and anaplerotic substrates in the TCA cycle [27]. Our analysis detected reduced levels of several amino acids in the CCC group, highlighting isoleucyl proline, lysine, citrulline, methionine, l-alanine, glutamine, asparagine, and arginine. On the other hand, only aspartate was significantly increased in CCC patients.

Currently, evidence from metabolomic analyses in heart failure (both in animal models and in human plasma samples) has yielded conflicting results. Studies using high-performance liquid chromatography in plasma samples from heart failure (HF) patients have reported significantly higher levels of most altered AAs compared to healthy controls [28]. These findings may be explained by an acceleration of protein breakdown, mainly in skeletal muscle [29]. Moreover, there is evidence of myocardial accumulation of AAs despite the increased demands for these molecules in the setting of HF, suggesting significant derangements in AA catabolism as was observed in a murine model of HF, in which a transcriptomic analysis revealed genes related to AA catabolism were downregulated [30]. On the other hand, the study by Aquilani R et al. reported a significant reduction in circulating AA concentrations in chronic HF compared to healthy controls, which was not explained by protein-calorie ingestion [31]. Lower levels of certain AAs might be attributed to increased consumption of these molecules by the myocardium, potentially as a compensatory mechanism in response to the energetic disbalance typically observed in the failing myocardium [32]. This AA depletion in the myocardium may finally lead to myocyte injury and cell death, favoring the progression of HF to more severe stages [33].

Regarding the clinical significance of AAs as biomarkers, the study of Hakuno et al. observed a significant association between monoethanolamine, methionine, tyrosine, 1-methylhistidine, and histidine concentrations in serum with echocardiographic parameters such as the left ventricle (LV) ejection fraction, LV end-diastolic volume index, inferior vena cava diameter, and the mean mitral E/e′ in HF patients [28]. Furthermore, a recently published cohort study observed that high 3-Me-His concentrations and low β-alanine and valine concentrations were independently associated with a significantly higher risk of a composite endpoint of all-cause death and readmission due to worsening HF or lethal arrhythmia in patients with HF. In addition, the inclusion of these amino acid levels in a multivariate model of clinical variables (including N-terminal proBNP levels) significantly improved its predictive capacity, highlighting the role of AAs as potential biomarkers for risk stratification in HF patients [34].

Finally, amino acid supplementation has been proposed as a promising therapeutic option in heart failure patients considering the significant alterations in the availability of AAs and their metabolic impact. A recently published systematic review and meta-analysis showed that essential AA supplementation was associated with significant improvements in relevant parameters such as six-minute walk test distance, muscle mass, and quality of life. Nevertheless, these findings should be interpreted with caution due to the limitations of those studies, such as the high risk of bias of the included trials and the high heterogeneity, which prevented performing a meta-analysis [35]. Further research is required to evaluate the changes in the activity of relevant pathways in the myocardium under AA supplementation to provide insights into the potential benefit of this therapy in the setting of CCC.

### 3.2. Significance of Acylcarnitines and Glycerophospholipids Biosynthesis

Acylcarnitines (ACs) facilitate the uptake and transport of fatty acids (FA) (C2-C26), which can be utilized for energy production in the mitochondria [36]. The transport of FA from the cytosol into the mitochondria, where FA β-oxidation occurs, is primarily controlled by a carnitine-dependent transport system that involves the enzyme carnitine palmitoyl transferase-1 (CPT-1). CPT-1 is in turn inhibited by malonyl CoA and stimulated by L-carnitine; therefore, increased levels of L-carnitine or ACs are responsible for increased FA oxidation, while its deficiency results in impaired mitochondrial β-oxidation, decreased glucose oxidation, and accelerated cellular apoptosis [37]. Nevertheless, the impact of ACs goes beyond FA oxidation, with several studies suggesting the critical roles of these molecules in stress response, chronic inflammation, protein metabolism, and oxidative injury prevention [38,39,40,41].

In the current study, we detected significantly decreased levels of myocardial ACs (O-malonyl-L-carnitine, hydroxy isovaleroyl carnitine, tiglyl carnitine) and long-chain fatty acids in tissue samples from CCC patients compared to healthy controls. On the other hand, a single class of ACs, methyl malonyl carnitine, was found to be increased in the CCC group (*p*-adj = 0.0078). These findings are compatible with what has been observed in other studies performed in tissue samples of patients with ESHF of other etiologies and highlight the critical role of ACs and mitochondrial fatty acid oxidation in CCC [42,43,44]. Although the reason behind the decreased myocardial ACs levels is still unclear in the setting of HF, several hypotheses have been proposed, highlighting a reduced abundance of the plasmalemma carrier OCTN2, which is responsible for cellular carnitine uptake and whose abundance was observed to be reduced in ESHF patients in the study of Bedi Jr et al. [42]. Moreover, mutations in the OCTN2 gene have been associated with primary carnitine deficiency in patients with cardiac involvement [45].

In the setting of CD, evidence regarding the role of acylcarnitines and glycerophospholipids has been derived from acute and chronic mouse models of CD, our study being the first one to show similar results in humans [16,24,46]. Moreover, the studies of Lizardo et al. and Hossain et al. observed that CCC is associated with a significant reduction of acylcarnitine concentration in the mice myocardium, these levels being potentially influenced by dietary fat content [23]. Furthermore, the study of McCall LI et al. which performed an untargeted LC-MS metabolomic analysis of heart sections from infected and uninfected mice, showed a significant association between the levels of several carnitine family members (highlighting *m*/*z* 512.468 RT 276s) and fatal outcomes two weeks after infection [47]. Despite representing a different scenario, studies performed on patients with HF have suggested that the levels of ACs and their metabolites are significantly associated with clinically relevant outcomes such as HF hospitalization and mortality [48,49]. Therefore, ACs may become potential predictive biomarkers of metabolic disruption and inflexibility in the failing myocardium [50]. On the other hand, although there is no evidence regarding the benefit of AC supplementation in the setting of CCC, the study of Hossain et al. suggested that carnitine treatment could significantly reduce mortality in acute T. cruzi infection through a process of disease tolerance, rather than reducing the parasite burden or altering its distribution [46].

Finally, our metabolic profiling detected reduced levels of unsaturated (Palmitoleic) and saturated (PC 15:0/0:0) FAs in cardiac tissue from CCC patients compared to healthy controls. These findings concur with previous data showing a downregulated expression of first- and third-step myocardial enzymes of FA β-oxidation in patients with ESHF and those undergoing cardiac transplantation [51]. Additionally, reports based on human and rodent models show downregulation of myocardial fatty acid oxidation and accelerated glucose oxidation in ESHF [51].

### 3.3. Significance of Glycolysis

It is known that a feature common to most etiologies of HF is the switching in myocardial substrate preference from FAs to glucose [41]. In this setting, glycolysis acts as a link between the pentose phosphate pathway (PPP), the TCA cycle, and the hexosamine biosynthetic pathways. Our results, in which seven out of nine glycolytic pathway intermediates were detected, showed that fructose-6-phosphate and lactate were significantly decreased in CCC patients, while 2- and 3-phosphoglycerate were increased (Table 1 and Figure 5). These results may be consistent with an increased metabolic efflux from glycolysis to other glycolysis-coupled pathways such as PPP, as oxidants can rapidly induce the rerouting of glucose flux into oxidative PPP and that multiple cycling of carbon molecules in PPP potentially amplifies NADPH production (Figure 5). The PPP also plays a relevant role in the pathophysiology of HF, as several studies have reported an increase in the activity of enzymes related to this pathway in HF patients compared to healthy hearts [52,53,54]. Moreover, Vimercati C et al. observed that PPP inhibition in a canine model of HF enhanced cardiac oxygen consumption, glucose oxidation, and normalized oxidative stress [55].

Nevertheless, recent evidence has suggested a dual role of PPP in the pathophysiology of HF. The study of Badolia R et al. evaluated the activity of glucose metabolism accessory pathways in HF patients before and after LVAD-induced mechanical unloading. The authors of this study observed higher activities of glucose-6-phosphate-dehydrogenase (G6PD), transketolase (TKT), and transaldolase (TALDO) in the failing heart compared to healthy donors; however, significantly higher levels of the G6PD, TKT, and TALDO, along with significant reductions of sedoheptulose-7-phosphate and fructose-6-phosphate were observed in HF patients who responded positively after mechanical unloading (LVEF > 40% and a left ventricular end-diastolic diameter of <6 cm) compared to those who did not respond [52]. The reason behind this observation is not entirely clear, but the authors hypothesize that an increased oxidative PPP flux could increase DAG1-glycosylation at the expense of ribitol and CDP-ribitol, as the levels of these molecules were observed to be significantly reduced in responder patients compared to those who did not respond to LVAD unloading and DAG1-glycosylation activity has been linked to the maintenance of the cytoskeletal integrity [56,57,58,59].

### 3.4. The Impact on the Tricarboxylic Acid (TCA) Cycle

As expected, the metabolic alterations present in CCC are also involved the TCA cycle, as five intermediates (cis-aconitate, succinic acid, (S)-malate, fumarate, and 2- methyl citrate) were found to be diminished in CCC samples compared to healthy controls (Table 1 and Figure 5). These findings are consistent with the results of the study of West JA et al., which evaluated a murine model of dilated cardiomyopathy. In this model, decreased concentrations of TCA cycle intermediates, carnitine derivatives, and glycolysis intermediates were shown, suggesting an altered energy metabolism in the failing heart [60]. Furthermore, studies evaluating the noncanonical functions of the TCA cycle in the heart have suggested potential cardioprotective roles of some cycle intermediates such as succinate and fumarate under conditions of myocardial injury [61]. Therefore, therapies modifying these relevant TCA cycle intermediates have been proposed as potential therapies for conditions such as ischemia–reperfusion injury, nonetheless, the evidence of the potential benefit of these therapeutic alternatives in heart failure is scarce [62,63,64].

### 3.5. The Potential Impact of Age on Metabolomic Profiling

Aging is associated with multiple changes in the cardiac tissue, highlighting pathological myocardial remodeling and microvascular dysfunction, which has been linked to multiple mechanisms, including mitochondrial dysfunction, aberrant signaling of the mechanistic target of rapamycin (mTOR), oxidative stress, autophagy, neurohormonal activation, dysregulation of miRNAs, among others [65,66]. Despite the advances made in the understanding of the pathophysiological processes related to aging in the myocardium, few studies have evaluated the metabolomic profiles related to these processes. The age difference present between the two groups in our study makes the analysis of age-associated metabolomic changes in myocardial tissue mandatory, as it is necessary to differentiate those findings potentially related to aging from those possibly explained by CCC.

At present, studies that have evaluated age-related changes in metabolomic profiles in cardiac tissue have been limited to murine models. These have reported significant age-related changes such as alterations in beta-oxidation mechanisms, with significantly lower levels of acylcarnitines, dysregulation of the TCA cycle, and a higher abundance of proteins involved in the process of glycolysis/gluconeogenesis such as glucose-6-phosphate and fructose-6-phosphate [67,68]. Nevertheless, our results report additional differences between the two groups not reported previously to be influenced by age, highlighting the increased concentrations of aspartate and the lower concentrations of BCAAs, as reports in the literature have suggested higher levels of BCAAs with increasing age, leading to activation of mTOR signaling and resulting in some of the structural and functional alterations observed in the aging myocardium [67,69,70,71,72]. On the other hand, the altered substrates in the TCA cycle in the present study have not been reported in other metabolomic profiling analyses related to aging. For example, in the study by Czibik et al., hearts of older mice were observed to have higher levels of fumarate compared with younger mice, potentially secondary to increased catabolism of phenylalanine at the cardiac level, which is contrary to the finding in the present study of lower levels of fumarate in the hearts of CCC patients [73]. Therefore, we highlight the potential differential profile of myocardial involvement secondary to chronic *T. cruzi* infection, which must be validated in future studies using matched cohorts.

### 3.6. Strengths and Limitations

This study represents the first metabolomic profiling analysis performed on human tissue samples from patients with CCC, a relevant achievement considering the multiple limitations in the availability and logistics for obtaining human heart tissue samples. Availability of the infrastructure and equipment necessary for the immediate collection and optimal storage of these clinical samples ensures the quality of the primary material, and thus the fidelity of the derived metabolic profiles. Finally, we highlight the inclusion of only identified metabolites (MSI category 1) in the pathways analyses, which increased the robustness of our data interpretation.

On the other hand, several limitations of our study deserve further discussion, highlighting the small number of patients that were enrolled in each group; therefore, we cannot rule out the presence of spectrum bias, and we highlight the need for larger, prospective validation studies to verify our results. The metabolites and associated pathways identified in this exploratory study serve as baseline for designing targeted metabolomic analyses, considering the highly significant *p*-values observed, which may suggest a relevant role of these metabolites in the pathophysiology of the disease. Moreover, we used samples from deceased patients as a control group, given the absence of discarded hearts from potential donors who did not meet the criteria required for use in cardiac transplantation. However, we emphasize that sample collection was as rapid as possible, being performed within the first hour after death. On the other hand, previous studies suggest that it is possible to use this type of sample in the absence of sources equivalent to the case group [74,75]. Furthermore, the significantly older age of patients with CCC represents a major limitation, as there is evidence of significant metabolic changes in the myocardium related exclusively to age; therefore, the results should be analyzed with caution. However, we highlight in the discussion that many of the metabolic pathways identified in the present study have been observed to be differentially altered in age-matched studies assessing other etiologies of HF and murine models of CD and CCC. Future studies comparing patients with ESHF of CCC etiology vs. other etiologies will allow identification of pathophysiological pathways unique to CCC that may differentiate it from other HF etiologies.

## 4. Materials and Methods

### 4.1. Chemical Reagents and Chromatography Column

LC-MS grade solvents and mobile phase additives were obtained from Sigma Aldrich (Gillingham, Dorset, UK). The ZIC-pHILIC column was obtained from VWR (Radnor, PA, USA).

### 4.2. Explanted Hearts

Tissue from the left ventricular free wall was obtained from explanted hearts of patients with end-stage heart failure (ESHF) secondary to CCC (*n* = 7) at the Fundación Cardiovascular de Colombia (FCV), a fourth-level cardiovascular referral center located in Floridablanca, Colombia. All patients were classified according to the New York Heart Association (NYHA) and the American Heart Association (AHA) criteria and were receiving medical therapy following the guidelines of the European Society of Cardiology.

Relevant information was extracted from the institutional medical record at the time of tissue collection, including sociodemographic and clinical data (age, diagnoses, medical therapy), hemodynamic monitoring, electrocardiogram, and doppler echocardiography. Echocardiograms were routinely performed according to the FCV management algorithms for patients listed for heart transplantation. Left ventricular size and function were quantified by visual inspection, tricuspid annular plane systolic excursion (TAPSE), the myocardial performance index (MPI), and the fractional area change (FAC) on the last echocardiogram performed before heart explantation.

Tissue from healthy hearts from unmatched organ donors (*n* = 7) who had died because of noncardiac causes (mainly stroke and traffic accidents) was used as the control group. These samples were obtained through the Institute of Legal Medicine and Forensic Sciences of Bucaramanga (Santander, Colombia). Briefly, sample collection was carried out by two professional teams: a forensic team and a medically trained team. They were on call 24 h a day. As soon as the teams were informed of a potential donor, they moved immediately to collect the samples and transport them to the Central Research Laboratory (CRL).

A pathologist and a microbiologist then immediately dissected and processed the hearts within 10 min while they were kept refrigerated on ice, following a rapid heart weight measurement and macroscopic examination. The pathologist led the systematic dissection while the other processed the collected tissues. The overall integrity of the excised native heart was maintained to obtain a conventional pathological evaluation. It should be noted that the time elapsed between death (healthy group), or explant (diseased group) and organ harvesting was no more than one hour. Once the samples arrived at the laboratory, they were processed under the same conditions. None of these donors showed evidence of heart disease and all presented normal left ventricular function and no history of myocardial disease or active infection at the time of death.

### 4.3. Tissue Preparation

Tissue explants were collected immediately following resection. Both CCC and control hearts were flushed with ice-cold before explant. At the time of explantation, two sections of the myocardial left ventricle free wall were collected from each participant. Approximately 100 mg of the frozen tissue per sample was mechanically homogenized and stored at −80 °C for further processing. Another tissue section was collected and fixed in 10% neutral buffered formalin and embedded in paraffin for histopathology studies.

### 4.4. Extraction of Metabolites from Heart Tissue

Metabolites were extracted using the methanol/chloroform method described by Le Belle et al. (2002) [8]. Briefly, 100 mg of frozen cardiac tissue was ground on dry ice using a mortar and pestle and placed inside 2 mL screw-capped flat-bottom tubes to which 600 μL of ice-cold 2:1 methanol:chloroform was added. Tissue samples were lysed using zirconium beads in a tissue lyser (Qiagen, Hilden, Germany) for 10 min at 25 Hz to ensure optimal extraction. Water and chloroform (200 μL each) were added, and the samples were thoroughly vortexed before centrifugation at 13,200 rpm for 25 min. After centrifugation the aqueous (top layer) and organic (bottom layer) fractions were separated and aliquoted into separate tubes and were stored at −80 °C before analysis.

#### LC/MS Analytical Platform and Methods

Hydrophilic interaction liquid chromatography (HILIC) was carried out on a Dionex UltiMate 3000 UHPLC system (Thermo Fisher Scientific, Hemel Hempstead, UK) using a ZIC-pHILIC column (150 × 4.6 mm, 5 μm column, Merck Sequant). Chromatography solvents consisted of 20 mM ammonium carbonate in water (A) and acetonitrile (B). The column was maintained at 30 °C and a gradient program was used as follows: time 0 min: 20% A and 80% B, 30 min: 80% A and 20% B, 31–36 min: 92% A and 8% B, 37 min: 20% A and 80% B. The flow rate was 0.3 mL/min, the injection volume was 10 μL and samples were maintained at 5 °C before injection.

For the MS analysis, samples were analyzed on a Q-Exactive mass spectrometer (MS) (Thermo Fisher Scientific) at Glasgow Polyomics, University of Glasgow. The MS was operated in polarity switching mode and the settings were as follows: resolution 70,000, AGC 1e6 (full scan mode), *m*/*z* range 70–1050, electrospray probe temperature 150 °C, and capillary temperature 320 °C.

The calibration mass range was extended to cover small metabolites by the inclusion of low-mass calibrants with the standard Thermo calmix masses (below *m*/*z* 138), butylamine (C4H11N1) for positive ion electrospray ionization (PIESI) mode (*m*/*z* 74.096426) and COF3 for negative ion electrospray ionization (NIESI) mode (*m*/*z* 84.9906726). A standard mix containing approximately 200 metabolites was run at the start of every analysis batch to aid metabolite identification as described by Creek DJ, et al. [9]. All samples from the experiment were analyzed in the same analytical batch and the quality of chromatography and signal reproducibility was checked by analysis of quality control samples, internal standards, and total ion chromatograms. A pooled quality control (QC) sample was generated using an aliquot of each biological sample and this QC sample was run throughout the batch every 6th sample and allowed to monitor the stability and quality of the LC-MS run. Samples were run in a randomized order.

### 4.5. Data Acquisition and Processing

All raw files were converted into mzXML format, thereby centroiding the mass spectra and separating positive and negative ionization mode spectra into two different mzXML files using the command line version of MS convert (ProteoWizard). Accurate masses of standards were obtained within 3 ppm accuracy and intensities of the quality control samples were within specifications. The raw data were preprocessed and analyzed using Polyomics integrated Metabolomics Pipeline (PiMP) (http://polyomics.mvls.gla.ac.uk (accessed on 5 June 2017) software platform [76]. The preprocessing strategy involved peak detection, peak matching, and retention time alignment. A list of peaks was obtained in which each peak is represented by its *m*/*z* value, retention time, and intensities across samples. A pooled quality control (QC) sample was analyzed to correct signal drifts and remove batch effects, as the basis for performing drift correction.

#### Metabolite Identification and Pathway Analysis

The chemical structures of the candidate metabolites were determined by comparing the retention times and mass spectra to the commercial standards. The accurate mass and structure information of candidate metabolites were also matched with those of metabolites obtained from HMDB (www.hmdb.ca (accessed on 5 June 2017) [77], and METLIN (metlin.scripps.edu/ (accessed on 5 June 2017) databases [78]. Metabolite identifications were given at level 2 according to the Metabolomics Standards Initiative (MSI) where accurate masses and predicted retention times were used to yield putative annotations, and these were termed “annotated metabolites”; when retention times of authentic standards were available, the identifications were considered as level 1, and these termed “identified metabolites” [19]. For the metabolites that significantly changed between tissues from CCC and healthy donors (*p* < 0.05), a metabolite set enrichment analysis (MSEA) and metabolic pathway analysis (PA) were performed using the MetaboAnalyst^®^ web portal and the KEGG database [79,80].

For the MSEA an Over Representation Analysis (ORA) algorithm was applied using a hypergeometric test [81]. MSEA is an approach to determine whether known biological functions or processes are over-represented (=enriched) in an experimentally derived metabolite list. A PA was also conducted, which combines the results of pathway enrichment analysis with pathway topology to aid in the identification of the most relevant pathways involved in the conditions of the study [82]. These two analysis tools were used in the identification of pathways to discuss in connection with significant metabolites.

### 4.6. Statistical Analysis

Categorical data are presented as absolute numbers and proportions, and continuous variables are expressed in terms of medians and interquartile ranges. For the metabolomics statistical analyses, a matrix including the peak number, sample name, and the normalized peak intensity was analyzed by principal component analysis (PCA) using PiMP and R software. Differentially abundant metabolites were defined based on both *p* values from *t*-test and log2 fold change (FC) values using PiMP software: differential abundance threshold log2FC > 1 < and a *p*-value < 0.05. Whenever a compound level was below the level of detection in some samples (missing values) and not in others, an imputed value equal to the minimum observed among all the samples of the study was included. The heatmap of different metabolites was performed using a MetaboAnalyst platform (http://www.metaboanalyst.ca (accessed on 5 June 2017) and the normalization procedure consisted of mean-centering and division by the standard deviation of each variable (Appendix A).

### 4.7. Ethics Statement

The Ethics Committees of Universidad Industrial de Santander (UIS) and the Fundación Cardiovascular de Colombia (FCV) approved the study protocol (institutional codes: D14-10586 and 298, respectively). In all cases, CCC patients or relatives of the healthy donors provided written informed consent. All clinical investigations were conducted according to the principles expressed in the Helsinki Declaration and its most recent amendment [83].

## 5. Conclusions

The results of the present exploratory study suggest an important role of metabolic alterations in the pathophysiology of CCC, highlighting altered branched-chain amino acid and acylcarnitine levels, reduced fatty acid uptake and oxidation, activation of other glycolysis-coupled pathways such as PPP, and dysregulation of the TCA cycle. Further studies involving larger sample sizes and comparisons with age-matched heart failure patients without CCC are needed to validate these results, potentially opening an avenue to investigate new therapeutic approaches for the treatment and prevention of the progression of this unique and severe cardiomyopathy. One of the areas of interest for future studies should include improving the availability of energy substrates through the use of metabolic modulators, which could potentially be beneficial for patients with CCC, even in cases of advanced heart failure.

## Figures and Tables

**Figure 1 ijms-23-10456-f001:**
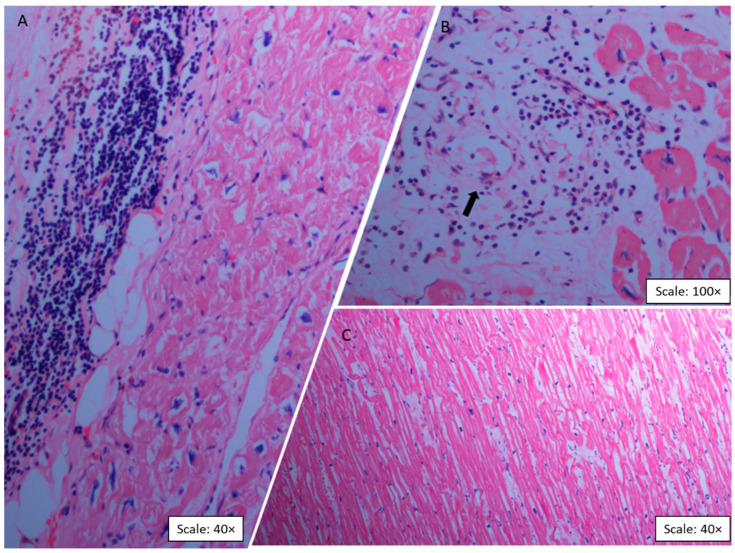
Representative histopathological findings on cardiac tissue sections with hematoxylin–eosin staining. Panels A and B correspond to cardiac muscle sections from patients with CCC. (**A**) Intense interstitial and perivascular fibrosis, inflammatory infiltrate, and myocytolysis are observed. The inflammation, composed of numerous macrophages, lymphocytes, plasma cells, and fewer eosinophils, surrounds degenerated and necrotic cardiomyocytes (Scale: 40×). (**B**) Abundant mononuclear cell infiltration in interstitial tissue and necrosis of cardiomyocytes (Scale: 100×). The arrow indicates the presence of a mononuclear inflammatory infiltrate with perivascular and interstitial distribution. (**C**) Control group sample showing heart tissue with a normal histological structure (Scale: 40×).

**Figure 2 ijms-23-10456-f002:**
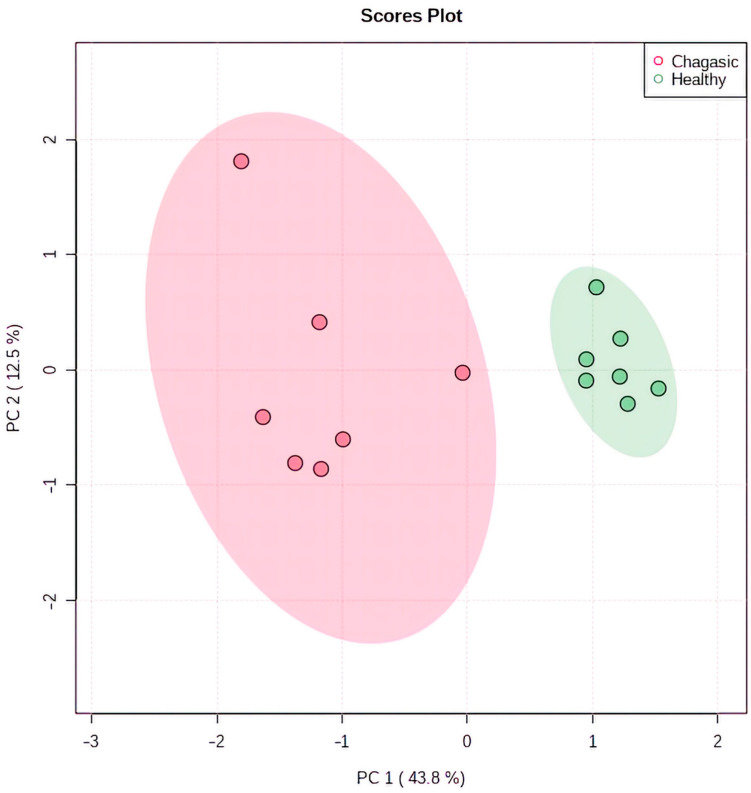
PCA analysis of LC/MS spectral data of cardiac tissue samples from CCC and healthy controls. Ellipses represent mean ± SD for each group.

**Figure 3 ijms-23-10456-f003:**
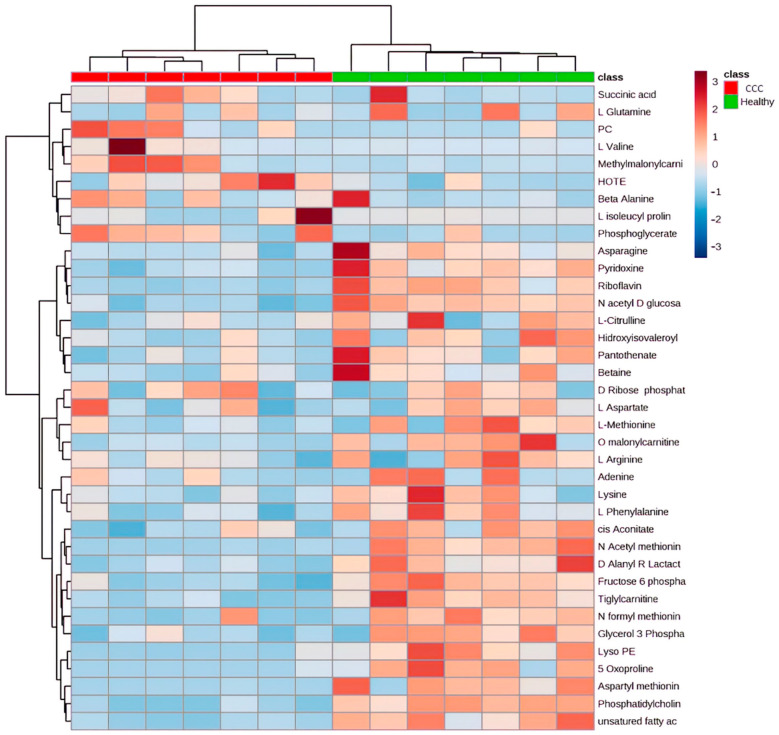
Hierarchical Clustering Heatmap of significantly and differentially abundant metabolites between CCC and Healthy donors. Columns correspond to samples and rows to individual metabolites. The color scale indicates the relative abundance of metabolites: red being the most abundant and blue the less abundant metabolites.

**Figure 4 ijms-23-10456-f004:**
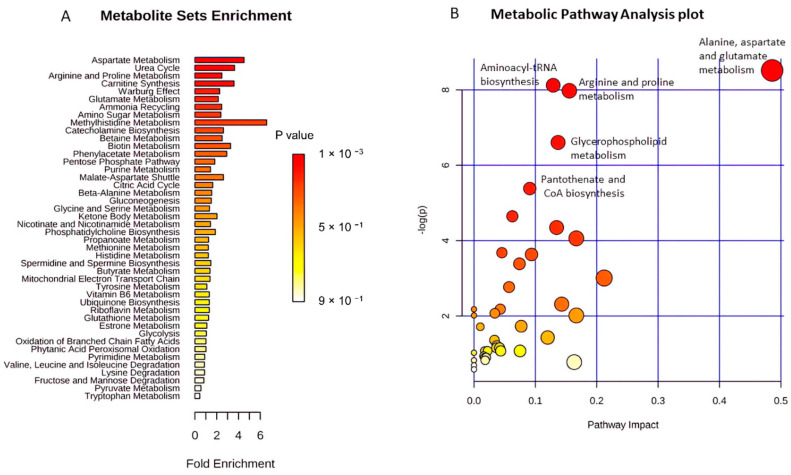
Metabolic pathways enriched in CCC patients compared to healthy controls. (**A**) Metabolic pathways enriched ordered by statistical significance, (**B**) functional analysis depicting enriched metabolic pathways according to pathway impact score.

**Figure 5 ijms-23-10456-f005:**
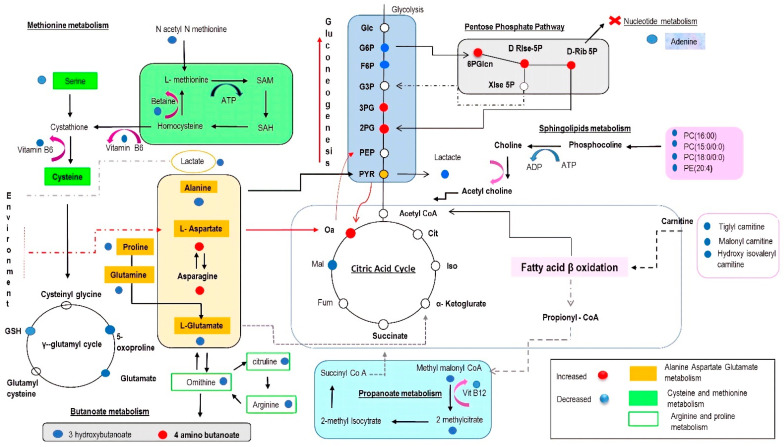
Schematic representation of significantly altered metabolites and metabolic pathways affected in CCC heart tissue vs. healthy donors. Diagram representing the level of metabolites in cardiac tissue from patients with CCC. Blue-colored spots represent decreased levels of metabolite. Red-colored spots represent an increased metabolite level in CCC while the white spots represent metabolites that did not show changes between the groups.

**Table 1 ijms-23-10456-t001:** Identity of metabolites with significantly different abundance between CCC patients and healthy donors.

Metabolite Number	ID Peak	Identified Metabolites	Log2 Fold Change	m/Z	Retention Times (mins)	Chemical Class	*p*-Value	Adjusted *p*-Value *
1	128	N-Acetyl Methionine	−4.75	1.920.688	422.47	C7H13NO3S	0.0001	0.0004
2	559	Aspartyl Methionine	−4.37	263.07	475.91	C9H16N205S	<0.0001	0.0001
3	29	PC (15:0/0:0)	−3.85	504.307	240.79	C23H48NO7P	<0.0001	0.0001
4	436	Lyso PE (16:0/0:0)	−3.68	4.522.792	250.58	C21H44NO7P	<0.0001	0.0001
5	465	L-Noradrenaline	−3.53	170.081	464.47	C8H11NO3	<0.0001	0.0001
6	12	L-Isoleucyl l proline	−3.39	2.291.542	561.53	C11H20N2O3	<0.0001	0.0001
7	68	O-malonyl L-carnitine	−3.08	2.481.122	695.43	C10H17NO6	<0.0001	<0.0001
8	277	Riboflavin	−3.0	3.771.445	486.78	C17H20N4O6	<0.0001	<0.0001
9	34	5-Oxoproline	−2.88	1.300.498	785.71	C5H7NO3	<0.0001	<0.0001
10	213	N Formyl Methionine	−2.87	1.780.531	496.84	C6H11NO3S	<0.0001	<0.0001
11	45	Adenine	−2.50	1.360.617	546.22	C5H5N5	0.004	0.0128
12	92	Pirbuterol	−2.36	2.411.541	590.07	C12H20N2O3	<0.0001	<0.0001
13	192	N-Acetyl-D glucosamine	−2.34	222.097	657.25	C8H15NO6	<0.0001	0.0003
14	550	Fructose 6 phosphate	−2.12	2.580.383	795.46	C6H14NO8P	<0.0001	0.0003
15	163	Tiglylcarnitine	−2.09	2.441.538	464.78	C12H21NO4	0.0001	0.0006
16	300	Pyridoxine (B6)	−2.07	170.081	464.47	C8H11NO3	<0.0001	0.0003
17	23	PC (16:0/0:0)	−2.03	496.339	249.4	C24H50NO7P	<0.0001	0.0003
18	439	Succinic acid	−2.00	1.170.193	797.5	C4H6O4	0.002	0.007
19	455	Lysine	−1.92	1.450.983	1160.9	C6H14N2O2	0.0008	0.0033
20	69	Hidroxyisovareoyl carnitine	−1.90	2.621.643	587.18	C12H23NO5	0.002	0.0011
21	588	D-Alanyl R Lactate	−1.81	1.600.616	505.38	C6H11NO4	0.0004	0.0025
22	89	2 Methylcitrate	−1.80	2.070.505	1160.39	C7H10O7	0.0005	0.0021
23	388	Glycerol 3 phosphate	−1.70	1.710.066	744.30	C3H9O6P	0.0001	0.0005
24	75	L-Citrulline	−1.61	1.360.617	546.22	C6H13N3O3	0.0007	0.003
25	64	L-Methionine	−1.57	1.500.583	644.53	C5H11NO2S	0.0001	0.0005
26	60	PC (16:1(9Z)/18:1(11Z))	−1.49	7.585.677	213.42	C42H80NO8P	0.0346	0.0052
27	367	β-Alanine	−1.46	880.404	774.77	C3H7NO2	0.0011	0.0043
28	633	cis-Aconitate	−1.43	1.730.092	736.89	C6H6O6	0.0114	0.0334
29	360	L-Glutamine	−1.43	1.450.619	784.69	C5H10N2O3	0.0154	0.0375
30	21	D-Valine	−1.36	1.180.862	632.6	C5H11NO2	0.0001	0.0008
31	18	Betaine	−1.36	1.180.862	630.34	C5H11NO2	0.0003	0.0015
32	96	Asparagine	−1.31	1.330.605	797.42	C4H8N2O3	0.0013	0.0048
33	52	L-Arginine	−1.30	175.11	1249.8	C6H14N4O2	0.0006	0.0025
34	35	L-Phenylalanine	−1.19	1.660.861	581.32	C9H11NO2	0.0022	0.0075
35	73	4 Aminobutanoate	1.04	1.040.705	796.8	C4H9NO2	0.0004	0.0019
36	471	D-Ribose 5 phosphate	1.13	2.290.119	799.22	C5H11O8P	0.00032	0.0022
37	395	L-Aspartate	1.83	1.320.303	783.88	C4H7NO4	0.0031	0.0098
38	218	2 Phospho D glycerate	1.36	187.000	847.47	C3H707P	0.0073	0.0201
39	496	Methylmalonylcarnitine	2.49	260.114	686.2	C11H19NO6	0.0023	0.0078
40	39	Tranexamic acid	8.94	1.581.173	804.32	C8H15NO2	<0.0001	<0.0001

* Adjusted *p*-value using the Benjamini–Hochberg method.

## Data Availability

Data is contained within the article or Appendix A.

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
