# Peer review of "Metabolomic Profiling of End-Stage Heart Failure Secondary to Chronic Chagas Cardiomyopathy"

_ijms, 2022, doi:10.3390/ijms231810456_

Round 1

Reviewer 1 Report

Metabolomic studies based on human heart tissue specimens are of high significance for the diagnosis and treatment of CCC. This article uses 14 human samples to provide new ideas for the diagnosis and treatment of CCC through metabolomic studies, and is a relatively practical article. However, there are still some questions to be discussed before the article be published.

1. A proper data analysis should come from a more equivalent case and control group. However, in this paper, the information from the case and control groups seems to be poorly matched, for example, the large difference in age between the two groups, since we know that the heart, as one of the most energy-consuming organs, is correlated with physical status of the organism. The authors should clarify the interference that this difference may (or may not) have on the data analysis in the discussion.

2.The hearts  samples in the case group was explanted hearts, while control group was from forensic autopsy, respectively. I am not sure that the hearts removed in the survival state and the hearts after death are comparable; after all, death cannot be excluded as a confounding factor. And the cause of death in these healthy controls should be accounted for in more detail.

3. The case group has very complete case information, but the control group has essentially no data. Table 1 does not seem to serve any comparative things, and I suggest that these data could be briefly described in text.

4. The author has made a relatively complete discussion, but there are several points that need to be clarified: (1)Line 298 "On the other hand, although there is no evidence regarding the benefit of AAs supplementation in the setting of CCC, the study of Hossain et al. suggested......" This paragraph is supposed to discuss amino acid-related content, and the introduction of carnitine therapy here seems to be very abrupt. Perhaps the author believes that carnitine is also an amino acid, but that it provides energy in the form of fat breakdown. (2) It seems inappropriate to show figure6 in the discussion, which should be part of the results.

5. Finally, the picture definition of the whole text is poor, the font size is not uniform, some data overlap with each other (Figure 2), the labeling is chaotic (Figure 4), and the scale application is not standard (Figure 1), which all needs to be greatly improved.

Author Response

Reviewer 1:

  1. A proper data analysis should come from a more equivalent case and control group. However, in this paper, the information from the case and control groups seems to be poorly matched, for example, the large difference in age between the two groups, since we know that the heart, as one of the most energy-consuming organs, is correlated with physical status of the organism. The authors should clarify the interference that this difference may (or may not) have on the data analysis in the discussion.

Response: We thank the reviewer for the important comments and agree with the statements made. Age differences represent a critical limitation of our study and require special attention in the discussion. Therefore, we have added a new section in which we discuss the results of studies that have evaluated changes in metabolomic profiles with aging and make a contrast with the results observed in the present study. In this way we can more clearly suggest which differences are probably attributable to aging and which could be potentially attributable to chronic Chagas cardiomyopathy (Page 11, lines 370-401).

2.The hearts  samples in the case group was explanted hearts, while control group was from forensic autopsy, respectively. I am not sure that the hearts removed in the survival state and the hearts after death are comparable; after all, death cannot be excluded as a confounding factor. And the cause of death in these healthy controls should be accounted for in more detail.

Response: We understand the reviewer's concern and agree on the relevance of addressing the sources of sample collection in both groups. Unfortunately, we did not have access to discarded hearts from donors who did not meet the criteria for cardiac transplantation due to the logistical limitations of the network, so we decided to obtain controls from deceased patients. Nevertheless, we have described in the methodology detailed aspects of the sample collection process, highlighting the permanent availability of the samples collection team and that the samples were collected and processed in less than one hour after death in all cases (Page 13, lines 455-469). In the results section we mention that all the patients died of cranioencephalic trauma in the absence of thoracoabdominal trauma (Page 2, lines 93-94). In addition, we have included a section in the limitations section mentioning this aspect and citing studies that suggest the usefulness of this type of sample in the absence of other samples equivalent to the case group (Page 12, lines 417-422).

  1. The case group has very complete case information, but the control group has essentially no data. Table 1 does not seem to serve any comparative things, and I suggest that these data could be briefly described in text.

Response: We thank the reviewer for this suggestion, we have now deleted Table 1 and improved the description of the included sample in the first paragraph of the results section (Page 2, lines 82-94).

  1. The author has made a relatively complete discussion, but there are several points that need to be clarified: (1)Line 298 "On the other hand, although there is no evidence regarding the benefit of AAs supplementation in the setting of CCC, the study of Hossain et al. suggested......" This paragraph is supposed to discuss amino acid-related content, and the introduction of carnitine therapy here seems to be very abrupt. Perhaps the author believes that carnitine is also an amino acid, but that it provides energy in the form of fat breakdown. (2) It seems inappropriate to show figure6 in the discussion, which should be part of the results.

Response: We thank the reviewer for highlighting this aspect of the discussion. We also understand the comment about the relationship between carnitine and amino acids. While it is true that carnitine is not an amino acid, it is synthesized from lysine. Also, the paragraph in which it was included was the only one that discussed supplementation therapy, so we considered including this information there. However, we understand the possible confusion this may generate, so we have moved these lines to the end of the third paragraph in the section "3.2. Significance of acylcarnitines and glycerophospholipids biosynthesis ". Finally, we have moved figure 6 to the results as suggested (Page 10, lines 314-318).

  1. Finally, the picture definition of the whole text is poor, the font size is not uniform, some data overlap with each other (Figure 2), the labeling is chaotic (Figure 4), and the scale application is not standard (Figure 1), which all needs to be greatly improved.

Response: We thank the reviewer for highlighting areas for improvement in the figures. We have now added the respective scales in Figure 1, improved the quality of Figures 2, 4, 5 and 6, removed the superimposed text in Figure 2 and removed Figure 3 due to its low resolution. We have also improved the labeling of Figure 4.

Reviewer 2 Report

The manuscript by Diaz et al, "Metabolomic Profiling of End-Stage Heart Failure Secondary to Chronic Chagas Cardiomyopathy", was reviewed.

This is an exploratory study to assess the underlying pathogenic mechanisms in the failing myocardium of patients with end-stage heart failure secondary to chronic Chagas cardiomyopathy. Using an untargeted metabolomic profiling approach, metabolic changes in the pathogenesis of Chagas cardiomyopathy were demonstrated, including altered levels of branched-chain amino acids and acylcarnitine, reduced fatty acid uptake and oxidation, activation of the pentose phosphate pathway and dysregulation of the TCA cycle.

This metabolomic profiling may have a role in contributing to the development and maintenance of the chronic inflammatory state that leads to the progression and severity of chronic Chagas cardiomyopathy.

Unfortunately, however, there are several problems as it is.

1. Table 1 LVEF (%) 10[10-15] Is it correct?

2. Fig. 1 Please show the scale of each panel. What does the arrow mean?

3. Fig 5 and Fig 6 are hard to read. Please write them in larger letters.

Author Response

Reviewer 2:

  1. Table 1 LVEF (%) 10[10-15] Is it correct?

Response: Yes, it represents the median and quartile 1 and 3. We have clarified this meaning in paragraph 1 of the results section.

  1. Fig. 1 Please show the scale of each panel. What does the arrow mean?

Response: We have now included the scale in each panel and added the function of the arrow in Figure 1.

  1. Fig 5 and Fig 6 are hard to read. Please write them in larger letters.

Response: We have now improved the resolution of figures 5 and 6. The reviewer can also directly access the figure files, as it is known that they may lose quality when included in a Word document.

Round 2

Reviewer 1 Report

The manuscript has been sufficientlly improved, and the authors have carefully addressed all my remaining questions.